# The High Mobility Group A1 (HMGA1) Chromatin Architectural Factor Modulates Nuclear Stiffness in Breast Cancer Cells

**DOI:** 10.3390/ijms20112733

**Published:** 2019-06-04

**Authors:** Beatrice Senigagliesi, Carlotta Penzo, Luisa Ulloa Severino, Riccardo Maraspini, Sara Petrosino, Hernan Morales-Navarrete, Enrico Pobega, Elena Ambrosetti, Pietro Parisse, Silvia Pegoraro, Guidalberto Manfioletti, Loredana Casalis, Riccardo Sgarra

**Affiliations:** 1Department of Life Sciences, University of Trieste, 34127 Trieste, Italy; beatric.senigagliesi@gmail.com (B.S.); carlotta.penzo.cp@gmail.com (C.P.); sar.petrosino@gmail.com (S.P.); d.pobega@yahoo.it (E.P.); spegoraro@units.it (S.P.); manfiole@units.it (G.M.); 2Nano Innovation Laboratory, Elettra-Sincrotrone Trieste S.C.p.A., 34149 Trieste, Italy; luisa.ulloaseverino@gmail.com (L.U.S.); elena_ambro@yahoo.it (E.A.); pietro.parisse@gmail.com (P.P.); 3Max Planck Institute for Molecular Cell Biology and Genetics, 01307 Dresden, Germany; maraspin@mpi-cbg.de (R.M.); moralesn@mpi-cbg.de (H.M.-N.)

**Keywords:** HMGA1, histone H1, chromatin, cancer, nuclear stiffness, mass spectrometry, atomic force microscopy (AFM), Stimulated emission depletion (STED) microscopy

## Abstract

Plasticity is an essential condition for cancer cells to invade surrounding tissues. The nucleus is the most rigid cellular organelle and it undergoes substantial deformations to get through environmental constrictions. Nuclear stiffness mostly depends on the nuclear lamina and chromatin, which in turn might be affected by nuclear architectural proteins. Among these is the HMGA1 (High Mobility Group A1) protein, a factor that plays a causal role in neoplastic transformation and that is able to disentangle heterochromatic domains by H1 displacement. Here we made use of atomic force microscopy to analyze the stiffness of breast cancer cellular models in which we modulated HMGA1 expression to investigate its role in regulating nuclear plasticity. Since histone H1 is the main modulator of chromatin structure and HMGA1 is a well-established histone H1 competitor, we correlated HMGA1 expression and cellular stiffness with histone H1 expression level, post-translational modifications, and nuclear distribution. Our results showed that HMGA1 expression level correlates with nuclear stiffness, is associated to histone H1 phosphorylation status, and alters both histone H1 chromatin distribution and expression. These data suggest that HMGA1 might promote chromatin relaxation through a histone H1-mediated mechanism strongly impacting on the invasiveness of cancer cells.

## 1. Introduction

Despite great progresses achieved in the understanding of cancer biology, metastases are still synonymous of terminal illness in several cancer types [1]. Among other aspects, the bidirectional interactions between cancer cells and the extracellular matrix (ECM) are necessary to sustain the process of metastatization. Cells can sense biochemical/biophysical properties of the ECM through their mechanosensing chain, which is constituted by focal adhesions, the cytoskeleton, the lamin, and the chromatin. External forces can thus propagate inside the cell up to the nucleus affecting its structure and influencing the modulation of the expression of specific set of genes, as a mechanism to adapt to environmental changes [2]. During invasion, cancer cells need to break through extracellular matrix openings that are smaller than their own diameters. In order to efficiently migrate in this crowded environment, cells adopt two, not mutually exclusive, strategies: cells can digest the surrounding ECM by secreting proteolytic enzymes, such as matrix metalloproteinases (MMPs) and/or they can become softer in order to undergo substantial deformations. Indeed, several studies have demonstrated that metastatic tumor cells are softer than non-malignant cells [3]. The nucleus is 2–10 times stiffer than cytoplasm, thus representing the more rigid physical barrier within the mechanosensing chain [4]. Nuclear stiffness is related to two strictly interconnected components of the nucleus: the nucleoskeleton, mainly composed of nuclear lamina [5] and the chromatin structure [6]. The nuclear lamina is a filamentous meshwork of lamin proteins juxtaposed to the inner membrane of the nuclear envelope that provides nuclear shape, supports its structure and contributes to organize the chromatin into distinct functional domains [7]. On the other side, the chromatin contribution to nuclear stiffness is mainly due to the degree of DNA compaction that is in turn linked to nucleosomal histone modifications and histone H1 [8,9]. Non-histone proteins also contribute to nuclear sturdiness influencing chromatin compaction state. For instance, HMGN5 (High Mobility Group N5) competing with histone H1 promotes chromatin decondensation [10] and impairs lamina organization thus strongly impacting the mechanical properties of nuclei [11]. The HMGA (High Mobility Group A) protein family belongs to the same High Mobility Group superfamily of HMGN proteins [12]. Despite distinct families differing in functions and structural domains, all HMG proteins lack any obvious specificity for DNA consensus sequences and compete with different mechanisms with histone H1 for DNA binding, thus negatively influencing the formation of higher order chromatin structures [13]. One peculiarity of HMGA proteins is that their expression is very high in embryonic [14,15] and tumor cells [16,17]. Importantly, HMGA proteins have been demonstrated to have a causal role in cancer onset and development [17]. HMGA proteins are encoded by two genes, HMGA1 and HMGA2, that give rise to three main proteins (HMGA1a and HMGA1b, two splicing variants of the HMGA1 gene, that from now on we will refer for simplicity as HMGA1, and HMGA2 from the HMGA2 gene). They are natively disordered architectural transcription factors [18] that contribute to the transcriptional regulation of several genes [19,20,21,22,23] in a context-dependent way [24] through the interaction with many different protein partners [25,26]. HMGA proteins have been demonstrated to be key actors in the process of epithelial-to-mesenchymal transition (EMT), cell motility, and invasion [19,27,28,29]. Here, we provide evidences that in breast cancer cells HMGA1 affects nuclear stiffness and that this effect could be, at least in part, explained through a mechanism involving histone H1, a protein linked to higher-order chromatin compaction. This work thus suggests that HMGA1 could exert its oncogenic activities also by modulating the biophysical properties of cells.

## 2. Results

### 2.1. Reversion of the Mesenchymal Phenotype in Triple Negative Breast Cancer (TNBC) Cells Causes an Increase in Cellular Stiffness

Metastatic tumor cells must be softer than non-malignant tumor and healthy cells to efficiently move through the extracellular environment [3]. We made use of the highly aggressive MDA-MB-231 TNBC cell line to test whether a reversion of the tumoral phenotype was associated with an increase of cellular stiffness. These cells are classified as mesenchymal-like according to their gene signature profile [30] because they express genes related to EMT, cell motility, and growth signaling pathways. We took advantage of the kinase inhibitor BI-D1870, which targets p90 RSK (ribosomal S6 kinase) isoforms (RSK1-4) [31], to induce a mesenchymal-to-epithelial transition (MET) in MDA-MB-231 cells since members of this kinase family were demonstrated to be involved in EMT [32] and to be particularly relevant for the proliferation of TNBC tumor-initiating cells [33]. The cell morphology (Appendix A) of MDA-MB-231 was not significantly affected after 24 h of DMSO treatment; on the contrary, as expected, after 24 h of BI-D1870 treatment, the morphology of MDA-MB-231 cells drastically changed from a spindle-like fibroblastic phenotype towards a flattened and polygonal morphology, a typical effect observed in MET (Appendix A). The stiffness of MDA-MB-231 cells treated with BI-D1870 was measured by atomic force microscopy (AFM) and compared to vehicle-treated (i.e., DMSO) cells. Cells treated with BI-D1870 display a higher cellular stiffness with respect to the control cell population as reported in Figure 1A and in the median and quantile distributions of Figure 1B. Therefore, when MDA-MB-231 cells revert to an epithelial morphology their stiffness is increased.

### 2.2. Modulation of HMGA1 Expression Levels Alters Cellular Stiffness in Breast Cancer Cell Lines

The expression of HMGA1 was shown to sustain the mesenchymal phenotype in TNBC cells [19,22]. We previously reported that HMGA1 orchestrates the expression of a plethora of factors involved in cell motility, invasion, metastasis, and stemness [19,20,21,34]. Given that HMGA1 is an essential chromatin structure modulator, we asked whether it could have a biophysical impact on cellular stiffness as well. To this end we silenced the expression of HMGA1 with siA1_3 [19] in the mesenchymal-like TNBC MDA-MB-231 and MDA-MB-157 cell lines, which express high level of this protein. We performed also the reverse experiment by using a previously established cell line [35] where HMGA1 is overexpressed in the Luminal A breast cancer cell line MCF7, where endogenous HMGA1 is barely detectable and cells exhibit an epithelial phenotype. In all these three cell lines, HMGA1 expression has been associated to the acquisition of a mesenchymal phenotype [19,36]. Western blot analyses showed that the modulation of HMGA1 expression levels has been obtained in the three cellular models as expected (Figure 2A). When the expression of HMGA1 is downregulated in aggressive mesenchymal tumor cells (i.e., in MDA-MB-231 and MDA-MB-157), cells became stiffer, while the opposite occurs when HMGA1 is overexpressed in epithelial MCF7 cells (Figure 2B,C).

### 2.3. HMGA1 Expression Is Linked to Histone H1 Phosphorylation Level

Nuclear stiffness partially depends on chromatin compaction [37]. The HMGA1 protein binds nucleosomes and DNA [24], it preferentially localizes in heterochromatin, and its distribution overlaps with that of histone H1 [38], one of the major determinants of DNA compaction [39]. It is worthwhile to evidence that the DNA binding properties of histone H1 are modulated both by competition with HMG proteins [13,40] and by its post-translational modifications (PTMs), above all phosphorylation [41]. Therefore, considering all these pieces of information we decided to evaluate whether HMGA1 could modulate nuclear stiffness via a mechanism involving histone H1. Firstly, we looked at histone H1 PTMs in all the cell lines previously analyzed by AFM. We took advantage of perchloric acid extraction to selectively extract HMG proteins and all histone H1 variants [42] and we analyzed histone H1 PTMs by liquid chromatography mass spectrometry (LC-MS). In Figure 3 two representative total ion current chromatograms (TICs) obtained from mass spectrometry analyses of control and MDA-MB-231 cells silenced for HMGA1 expression (MDA-MB-231: CTRL and siA1_3) are reported. Elaboration of the TIC provides information about the proteins eluting across the chromatographic separation. Inspection of the m/z spectra of each chromatographic peak allows the obtainment of the identities of the corresponding proteins. The location within the TICs of HMGA1a and HMGA1b (the two splicing variants of the HMGA1 gene), HMGB, HMGN1, and HMGN2 proteins, and the histone H1 variants are indicated in the TICs (Figure 3) while results concerning histone H1 PTMs are reported in Figure 4.

On the left, reconstructed mass spectra of control and HMGA1-silenced cells (MDA-MB-231 and MDA-MB-157: siCTRL and siA1_3) or HMGA1-overexpressing cells (MCF7: CTRL and HA-A1) samples are reported. As shown, MDA-MB-231 and MDA-MB-157 cells express two histone H1 variants (histone H1.2 and histone H1.4) while MCF7 cells, in addition to these two variants, also express the H1.3 and H1.5 variants. Experimental molecular masses have been compared with theoretical ones to assign identity and PTMs. All histone H1 variants are fully mono-acetylated and no changes in the acetylation levels were detected, therefore no indication regarding acetylation has been reported. Each histone H1 variant is differentially phosphorylated and the number of phosphates is indicated near each mass peak (0P, 1P, 2P, 3P, or 4P). On the right part of Figure 4, for each histone H1 variant, the percentages of unphosphorylated forms (0P) with respect to phosphorylated forms (1P + 2P + 3P + 4P) are reported, both for control and treated samples. Molecular masses of histone H1 are reported in Appendix A. Either the silencing or the overexpression of HMGA1 affected histone H1 phosphorylation thus establishing a direct correlation between HMGA1 expression levels and histone H1 phosphorylation status.

Since the treatment with BI-D1870 resulted in an efficient reversion of the MDA-MB-231 cellular phenotype (Appendix A), we analyzed histone H1 post-translational status in BI-D1870 treated cells with respect to control (DMSO) ones (Appendix A). Histone H1 turned out to be fully dephosphorylated following this drug-induced MET. Altogether, these evidences confirm the link between cellular phenotype, nuclear stiffness, and histone H1 phosphorylation status.

### 2.4. HMGA1 Expression Is Linked Both to the Spatial Organization within Chromatin and Protein Expression Levels Of Histone H1

Given that upon HMGA1 silencing we observed both an increase in nuclear stiffness and a decrease in histone H1 phosphorylation levels in TNBC cells and exactly the opposite effect upon HMGA1 overexpression in MCF-7 cells, we hypothesized that these events could be linked to gross chromatin alterations.

It has been previously shown by stochastic optical reconstruction microscopy (STORM) analyses that nucleosomes in interphasic cells are organized in defined nanodomains and that the number of “aggregated” nucleosomes (i.e., nucleosome clutches) reflects chromatin compaction [43]. Histone H1 was found to be enriched in big nucleosomes clutches that correspond to compact chromatin regions: its nanoscale distribution was strongly perturbed by Trichostatin A (TSA) [43], a histone deacetylase inhibitor that, by altering the acetylation state of core histones, is responsible for chromatin decompaction. Therefore we took advantage of super-resolution stimulated emission depletion (STED) microscopy to observe at nanoscale resolution whether changes in the HMGA1 expression levels were linked to modifications in the spatial distribution of histone H1 within the chromatin environment. We have been able to visualize (Figure 5A) and measure (Figure 5B–D) histone H1 clusters in control and HMGA1 silenced MDA-MB-231 cells. Under the same imaging conditions, histone H1 cluster area was, on average, bigger in the HMGA1 silenced cells rather than in the controls (Figure 5B), suggesting therefore a more compacted chromatin state upon HMGA1 depletion. In the same way, clusters were more concentrated in the control cells (Figure 5C). We considered cluster density (number of clusters/μm^2^) instead of the total number of clusters since the former measure takes into consideration also the size of the nucleus, providing clear information regarding histone H1 cluster abundance. We are aware of the fact that the absolute changes in cluster area and density are modest, however it has to be considered that the values are single-cell measurements that in turn are linked to the HMGA1 silencing efficacy, which is obviously different in each single cell.

This could also partially explain the reason why values obtained from siCTRL-treated cells are less dispersed with respect to those obtained from siA1_3-treated ones. Figure 5D shows the mean intensity per cluster, and, surprisingly, clusters turned out to be dimmer in siA1_3-treated cells with respect to control ones. The lower florescence intensity of histone H1 clusters in HMGA1 silenced cells could be due either to a lower histone H1 protein levels, in analogy with the silencing of HMGB1 which was demonstrated to be linked to a general decrease in nucleosomal histones and histone H1 expression levels [44], or to a more compact chromatin state which makes it less accessible to antibodies, or to a combination of both. To clarify this point, we performed western blot analysis comparing the protein expression level of one of the two histone isoforms expressed by MDA-MB-231 cells (histone H1.2) between siA1_3- and siCTRL-treated cells. The results clearly showed that histone H1.2 protein levels were lowered in HMGA1-silenced cells (Figure 5E), a situation that mimics the data obtained in HMGB1 KO cells [44].

### 2.5. Histone H1 Variants Have Differential Prognostic Values

Histone H1 has always been described as a nuclear factor involved in the formation of higher-order chromatin structure, i.e., the 30 nm chromatin fiber, which has been generally associated with a closed and inaccessible DNA conformation [45]. HMGA1 and histone H1 are two well-established competitors. In particular, several in vitro and in vivo data suggested that HMGA1 and histone H1 compete for the same DNA sequences and structures [40,46,47,48,49]. Moreover, using fluorescence recovery after photobleaching (FRAP), it was demonstrated that, in living cells, the expression of HMGA1 significantly lowered the histone H1 residence time [13]. Therefore, it was counterintuitive to find out that HMGA1 expression levels were directly linked to those of histone H1.2 (Figure 5E). However, new roles for the different histone H1 variants are recently emerging. It has been demonstrated that, in breast cancer cells, the expression of histone H1.2 is linked to cell-cycle progression, that its depletion is responsible for a cell-cycle arrest in G1, and that histone H1.4 has a role in cell survival [50]. Moreover, it has been demonstrated that histone H1 does not always have a repressive function and that it can also be associated with the up-regulation of specific sets of genes [50]. Histone H1 expression also correlates with cell-migration and the expression of EMT-markers in colon cancer cells [51] and with prostate cancer cell proliferation [52].

To shed light on these unexpected evidences, we obtained Kaplan Maier survival curves (Overall Survival, Relapse-Free Survival, and Distant Metastasis-Free Survival) interrogating breast cancer gene expression datasets. As it is possible to observe from Figure 6 we considered the replication-independent histone H1.0 (H1F0) and the five replication-dependent histones: H1.1, H1.2, H1.3, H1.4, and H1.5 (HIST1H1A, HIST1H1C, HIST1H1D, HIST1H1E, and HIST1H1B, respectively). It is noteworthy to observe the different prognostic behaviors of these six genes/proteins. Histone H1.0 expression turned out to be a prognostic factor indicating a good trend and this was expected since it is involved in cell differentiation [53], while histone H1.2 and H1.4 expression turned out to indicate a poor prognosis. These data were substantially confirmed evaluating another dataset (Appendix A). Overall, these evidences are in line with findings suggesting that each histone H1 variant has its own specific function(s) [54].

## 3. Discussion

Taken together, our data highlighted a direct link between HMGA1 expression levels, cellular stiffness, and histone H1 phosphorylation, nanoscale localization, and expression levels. Several key points must be recalled in order to discuss the possible mechanisms underlying these evidences.
(1)HMGA1 and histone H1 are DNA-binding competitors [13,40];(2)HMGA1 and histone H1 exert opposite effects: HMGA1 contributes to chromatin decondensation whereas histone H1 is involved in chromatin compaction [13];(3)HMGA1 regulates the expression of factors involved in histone H1 phosphorylation (i.e., cyclin E2) [34];(4)HMGA1 is involved in the recruitment on chromatin of factors involved in histone H1 phosphorylation. HMGA1 binds the 7SK non-coding RNA that in turn is in complex with P-TEFp (i.e., CDK9/cyclinT1/T2) [55];(5)Phosphorylation decreases histone H1 DNA binding affinity and hence HMGA1 could compete more efficiently with histone H1 [41].


Considering these evidences, we can hypothesize that HMGA1 regulates chromatin accessibility with convergent and not mutually exclusive mechanisms. Traditionally, HMGA1 induced genome-wide changes in chromatin decondensation are associated to gene expression activation. Data presented in this work let us speculate that the global chromatin opening, which is associated with high HMGA1 expression level [56], might also be linked to an alteration of the biomechanical properties of the nucleus. This could confer to cancer (and probably also embryonic) cells an enhanced plasticity and motility with histone H1 playing a prominent role as well.

Cellular stiffness has been suggested as biomarker for cancer detection and staging [57,58]. Despite the relationship between cellular motility and stiffness still being intensively debated [3], we feel that the comprehension of all the molecular factors contributing to cell mechanical properties represents an invaluable advancement towards the integration of the fascinating field at the border line between molecular biology and biophysics.

The data reported in this work enlarge the information regarding HMGA1-histone H1 relationship making a more complex scenario than a simple competition towards common DNA targets. Indeed, not only HMGA1 competes with histone H1 for DNA binding [13], but it is also involved in the modulation of H1 PTMs (and hence DNA binding activities) and H1 protein expression level. Histone H1 de-phosphorylation has been correlated with the process of cellular differentiation [59] while, on the contrary, a high level of HMGA1 expression, has been linked to undifferentiated and stem cell properties [19,60] that can be roughly considered opposite and interconnected features.

An unexpected fact that we highlighted is that the protein expression level of histone H1.2 seems to be positively regulated by HMGA1. This fact is counterintuitive: one would have expected that upon HMGA1 silencing histone H1 protein expression levels would have remained the same and, being dephosphorylated and in the absence of a strong competitor, it would have easily contributed to chromatin compaction. All HMG proteins have been reported to be histone H1 competitors and to strongly affect its chromatin dynamics [40]. Similarly to what reported within this study, it has been demonstrated that in HMGB1 KO cells histone H1 is reduced, together with all nucleosomal histones [44]. Intriguingly, it was demonstrated that there was a disproportionate loss of nucleosomes, being the DNA sites with lower nucleosome occupancy the more affected regions, while those more occupied displayed an increase in their nucleosomal occupancy [44]. Extending this reasoning to histone H1, this might imply that histone H1 can also undergo such differential behavior, and that in this way certain chromatin zones could become even more compacted upon histone H1 depletion. In addition to this, the association between histone H1.2 (and to a lesser extent of histone H1.4) expression with a worst prognosis indicates a positive role of these variants in conferring cancer cells’ distinctive aggressive traits and therefore HMGA1 could exert its oncogenic role by modulating histone H1 expression as well.

In our opinion, a relevant aspect underscored by our data is the articulated transcriptional and post-transcriptional relationships between chromatin architectural proteins. HMGA1 simultaneously up-regulates histone H1 expression levels and favors its phosphorylation, thus decreasing histone H1 DNA binding affinities. More histone H1 with an overall lower DNA-binding affinity should imply a higher occupancy of histone H1 but with a higher exchange rate, i.e., a more genome-spread and dynamic behavior. It is relevant to evidence here that histone H1 has been shown to act in a “hit and run” fashion; in the mouse mammary tumor virus (MMTV) promoter histone H1 favors in an initial step a specific nucleosome positioning within MMTV promoter itself, in turn responsible for a synergistic effect of progesterone receptor (PR) and nuclear factor 1 (NF1) towards transcription [61]. Subsequently, after PR recruitment, histone H1 is phosphorylated, it detaches from the MMTV promoter leaving the PR and NF1 full access to their transcription consensus sites and allowing transcription to start [61].

In conclusion, the dissection of the molecular mechanisms linked to chromatin architecture could provide important details about the chromatin dynamics involved both in embryonic development and in the process of neoplastic transformation in which HMGA proteins are highly expressed and play key roles. This work sheds light on a novel and unexpected link between chromatin architectural proteins expression and biophysical properties of cells unveiling novel and unexpected functional relationships among chromatin architectural proteins, which deserve further clarification.

## 4. Materials and Methods

### 4.1. Cell Culture and Treatments

Cell lines were kindly provided by the laboratory of prof. G. Del Sal (Laboratorio Nazionale CIB (LNCIB) Area Science Park, Trieste, Italy). Cells were cultured as previously described [20,35]: MDA-MB-231 and MDA-MB-157 cells were grown in DMEM supplemented with 2 mM L-glutamine, 10% tetracycline free fetal bovine serum (Tet-free FBS, Euroclone, Pero, Italy), penicillin (100 U/mL, Euroclone), streptomycin (100 μg/mL, Euroclone). MCF7 cells were grown in Dulbecco’s MEM Nutrient Mix F12 (1:1) with 25 mM HEPES (DME/F12-HEPES, Euroclone) supplemented with 2 mM L-glutamine, 10% tetracycline free fetal bovine serum (Tet-free FBS, Euroclone), penicillin (100 U/mL, Euroclone), streptomycin (100 μg/mL, Euroclone), and 1X MEM non-essential amino acids (Sigma Aldrich, St. Louis, MO, USA).

Silencing experiments were carried out essentially as previously described [20]; briefly, cell lines were transfected with 100 nM siRNAs with LipofectamineTM RNAiMAX reagent (Invitrogen, Carlsbad, CA, USA) and collected after 72 h. siRNA (Eurofins) sequences: HMGA1 (A1_3): ACTGGAGAAGGAGGAAGAG; CTRL: ACAGTCGCGTTTGCGACTG.

HMGA1 expression vector was obtained by subcloning the PCR products of the coding regions of human HMGA1a cDNA into the pcDNA3-HA plasmid expression vector (Invitrogen). pcDNA3-HA empty vector was used as a control. Plasmids were transfected into MCF7 cells using the Fugene reagent (Roche, Mannheim, Germany) following the manufacturer’s instructions. Clone selection was obtained by adding 1.5 mg/mL G418 disulfate salt (Sigma Aldrich) to the culture medium. MCF7_CTRL and MCF7_HMGA1 pools were obtained by combining single clones [35].

MDA-MB-231 cells were treated with BI-D1870 (10 μM) for 24 h while control cells with the vehicle (DMSO). The stock solution of BI-D1870 (Axon MedChem, #1528) was 20 mM in DMSO. DMSO in the control experiments was 0.05%.

### 4.2. SDS PAGE and Western Blot Analyses

SDS PAGE and western blot analyses were performed essentially as previously described [19]. WB signal intensities were normalized to total proteins using densitometric analysis of Ponceau S stained membranes. Primary antibodies used were: α-HMGA1 (homemade) and α-H1.2 (ab181977, Abcam, Cambridge, UK). Secondary antibodies were: α-Rabbit IgG Peroxidase conjugate (#A0545, Sigma Aldrich, St. Louis, MO, USA) and α-Mouse IgG Peroxidase conjugate (#A9044, Sigma Aldrich, St. Louis, MO, USA). The α-HMGA1 antibody was produced by immunization of a rabbit with a recombinant HMGA1a protein lacking the C-terminal acidic domain. α-HMGA1 antibodies were affinity purified form serum using a resin (Affi-Gel 10 Gel #153-6046, Bio-Rad Laboratories, Hercules, CA, USA) covalently derivatized with HMGA1 recombinant protein. Recombinant proteins used for immunization and affinity purification were purified by reversed-phase HPLC and checked by mass spectrometry. Antibodies were eluted with a Glycine 0.2 M pH 2.8 solution and acidic pH was neutralized with a Tris 2 M pH 8 solution. Antibodies were stored at −20 °C in small aliquots.

### 4.3. Atomic Force Microscopy

Cells have been seeded on coverslips (22 mm Ø) and treated as indicated. Cells were washed with PBS three times and fixed with PFA 4% for 20 min. Cells were washed with PBS three times and conserved in PBS with penicillin/streptomycin. To visualize nuclei, cells were stained with Hoechst (Sigma Aldrich) in PBS for 10 min. All the AFM measurements have been obtained using similar cell culture confluences in order to avoid results affected by alteration due to changes of cell–cell communication, cell polarity, inhibition of growth, or cell overlapping. Cell stiffness was evaluated from AFM nano-indentation measurements performed in correspondence of cell nuclei with a tipless silicon nitride cantilever (elastic constant 0.03 N/m and nominal resonance frequency of 10 kHz, CSG11B- from NT-MDT) on which a silica microsphere of 20 μm diameter was manually glued. In nano-indentation measurements, the deflection of the cantilever was measured when the bead is pushed towards the sample. Cantilever deflection and z-piezo movements were detected at each indentation step and converted in a force-displacement curve by knowing the cantilever spring constant [62]. Measurements have been performed at room temperature in the liquid cell of a commercial NT-MDT Smena (NT-MDT, Zelenograd, Moscow, Russia) AFM combined with an inverted optical microscope (Inverted Research Microscope Eclipse Ti, Nikon, Nikon Corporation, Tokyo, Japan). Stiffness of the material under the tip load was determined fitting the data with a Hertzian model of surface indentation [63].

### 4.4. LC-MS Analyses

HMG and histone H1 were selectively extracted by 5% (*w*/*v*) perchloric acid as previously described [42]. LC-MS analyses were performed as previously described [18] using an mRP Hi Recovery 0.5 × 100 mm, 5 µm (Agilent, Santa Clara, CA, USA) column. In each LC/MS analysis molecular masses have been internally adjusted according to the ppm error of HMGN2 molecular mass obtained after mass spectra deconvolution. The percentages of unphosphorylated and phosphorylated histone H1 variants have been calculated on the base of reconstructed peak heights (intensity) essentially as previously described [42].

### 4.5. Immunofluorescence Analyses

Sample for immunofluorescence analyses were prepared essentially as previously described [34]. The primary antibody was α-H1 (Abcam, ab71594) and the secondary antibody was goat anti mouse-StarRed (STRED-1001, Abberior GmbH, Göttingen, Germany).

STED imaging was performed using a custom designed Abberior 775 3D-2 Color-STED system with 100×/1.4 NA oil Olympus objective. Abberior Star Red was imaged with a pulsed laser at 640 nm. The depletion laser was a Katana 775 nm pulsed laser. To reduce high frequency noise, STED images were filtered with a 2D or 3D Gaussian with a sigma of 0.8 pixels. All images were processed and analyzed using the custom-designed image analysis software MotionTracking [64], as previously described [65,66,67]. Briefly, the images were first smoothed with a Gaussian low-pass filter (sigma = 0.01 µm) and the background intensity was subtracted. The objects were segmented by fitting the image intensity by a sum of powered Lorenzian functions (i.e., model-based segmentation) [64]. The different features (e.g., area, total intensity, mean intensity) were calculated from the segmented objects. The boundary of the nuclei was automatically identified using a threshold-based segmentation.

The statistical significance analysis was performed using two-sided *>t*>-test assuming unequal variances in MATLAB2018b. The bar plots were generated using the customized function provided in https://de.mathworks.com/matlabcentral/fileexchange/26508-notboxplot.

### 4.6. Breast Cancer Data Sets and Survival Analysis

Overall survival, metastasis free survival, and relapse free survival were evaluated using two Kaplan-Meier plotter web tools: Kaplan–Meier Plotter (kmplot.com) [68] and Gene Expression-Based Outcome for Breast Cancer Online (GOBO) (http://co.bmc.lu.se/gobo/) [69]. The samples were split into two groups according to the quantile expressions of the selected proteins.

## Figures and Tables

**Figure 1 ijms-20-02733-f001:**
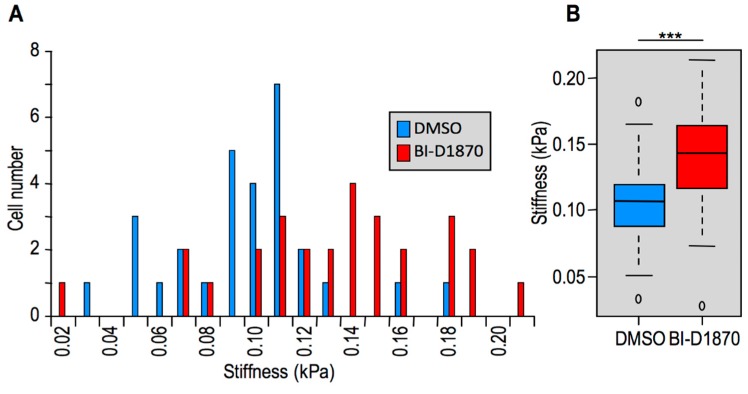
Cells undergoing mesenchymal-to-epithelial transition (MET) increase their stiffness. (**A**) Histograms graphs reporting stiffness distributions of the two cell populations analyzed (DMSO 0.05% and BI-D1870 10 μM). (**B**) Box plot illustrative of median and quantile distributions of the two cell populations (***: *p* < 0.001).

**Figure 2 ijms-20-02733-f002:**
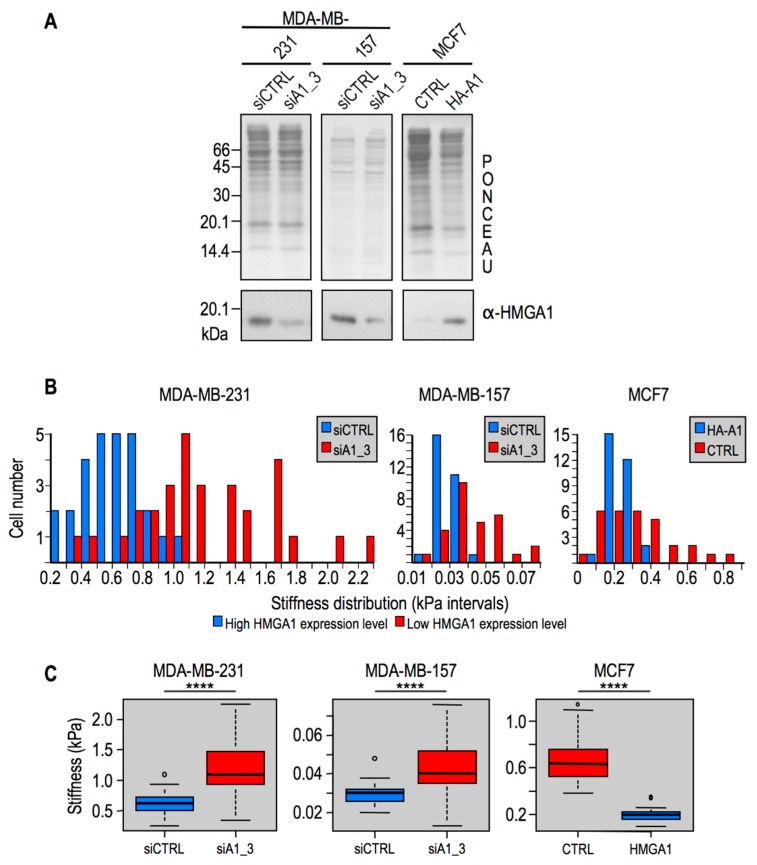
Cellular stiffness is modulated by changes in HMGA1 (High Mobility Group A 1) expression levels. In MDA-MB-231 and MDA-MB-157 cells HMGA1 expression has been silenced by siRNA, whereas in MCF7 cells HMGA1 has been overexpressed by means of transfection with a HA-HMGA1 protein expression vector. CTRLs indicate control experiments performed with siCTRL or an empty HA-expression vector. (**A**) Western blot analyses to assess HMGA1 protein expression levels in the three cellular models. Red ponceau membranes are shown as controls for protein loading normalization. Molecular weight markers are indicated on the left (kDa). (**B**) Stiffness distributions of all cell populations analyzed. (**C**) Box plots illustrative of median and quantile distribution of the three different cell population analyzed (****: *p* < 0.0001).

**Figure 3 ijms-20-02733-f003:**
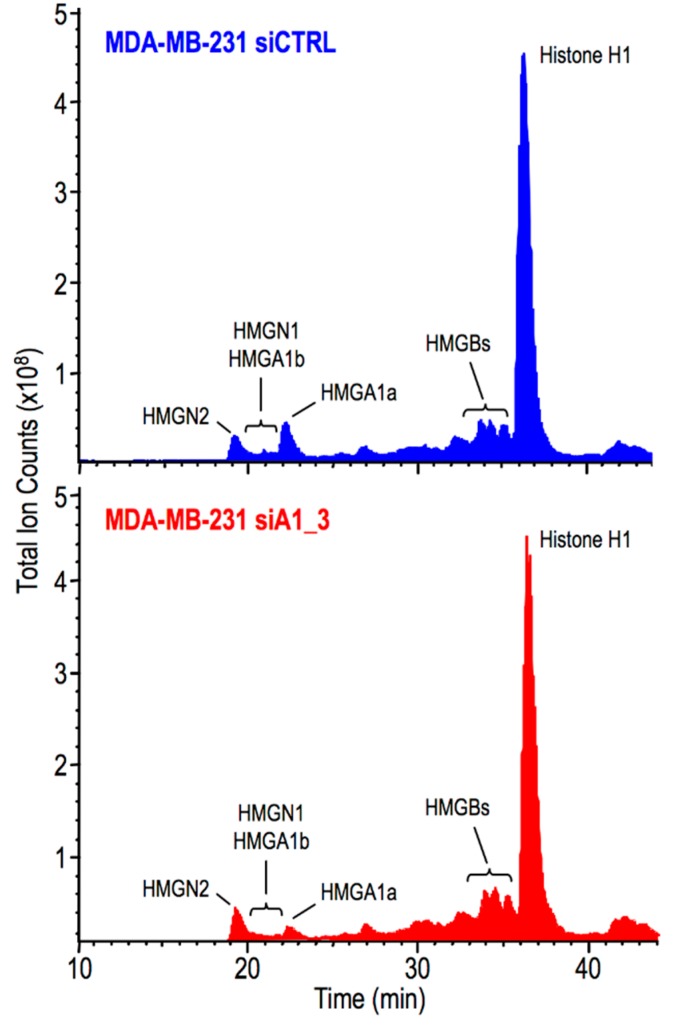
LC-MS analyses of HMGs and histone H1 variants. Comparison of total ion chromatograms (TICs) obtained by LC-MS analyses of MDA-MB-231 treated with siCTRL or siA1_3. The peaks within each chromatogram of HMGN2, HMGA1b, and HMGN1, HMGA1a, HMGBs, and histone H1 variants are indicated.

**Figure 4 ijms-20-02733-f004:**
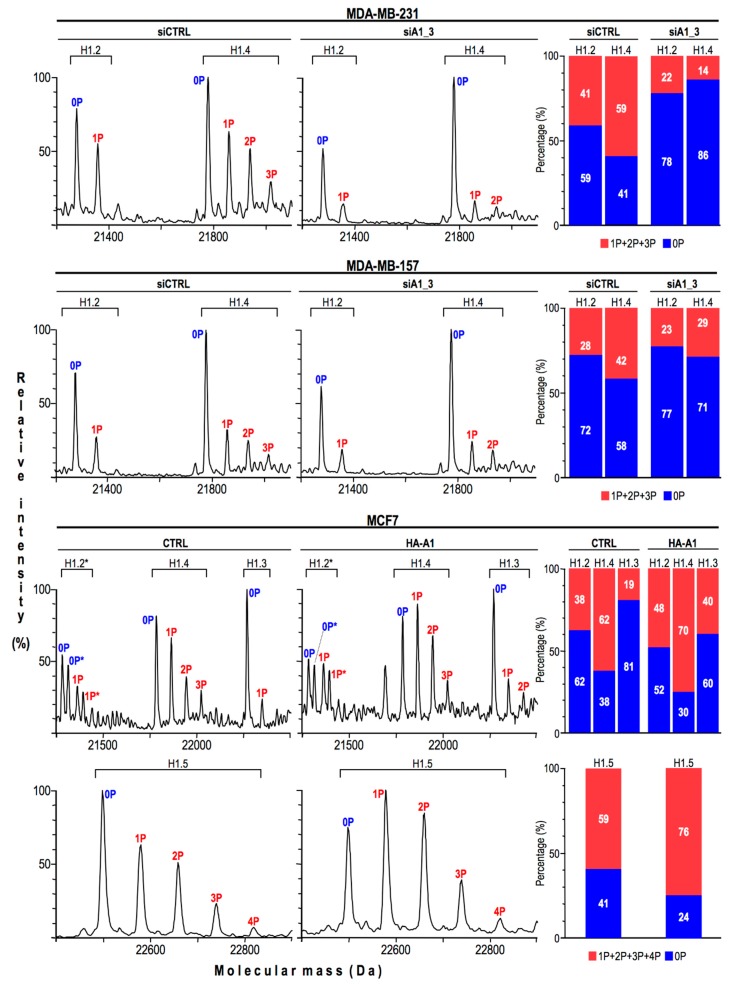
Histone H1 phosphorylation is linked to HMGA1 expression level. Left part: Reconstructed mass spectra of histone H1 variants obtained from MDA-MB-231 and MDA-MB-157 cells treated with siCTRL and siA1_3 and from control (CTRL) and HA-HMGA1 overexpressing (HA-A1) MCF7 cells. The various histone H1 variants are indicated above each spectrum. 0P, 1P, 2P, 3P, and 4P indicate the number of phosphate groups. *: MCF-7 cells express two histone H1.2 allelic variants. Right part: Histogram graphs showing for each histone H1 variant the distribution in terms of percentage between unphosphorylated (0P) and phosphorylated forms (1P+2P+3P+4P). The percentage values are indicated within each bar.

**Figure 5 ijms-20-02733-f005:**
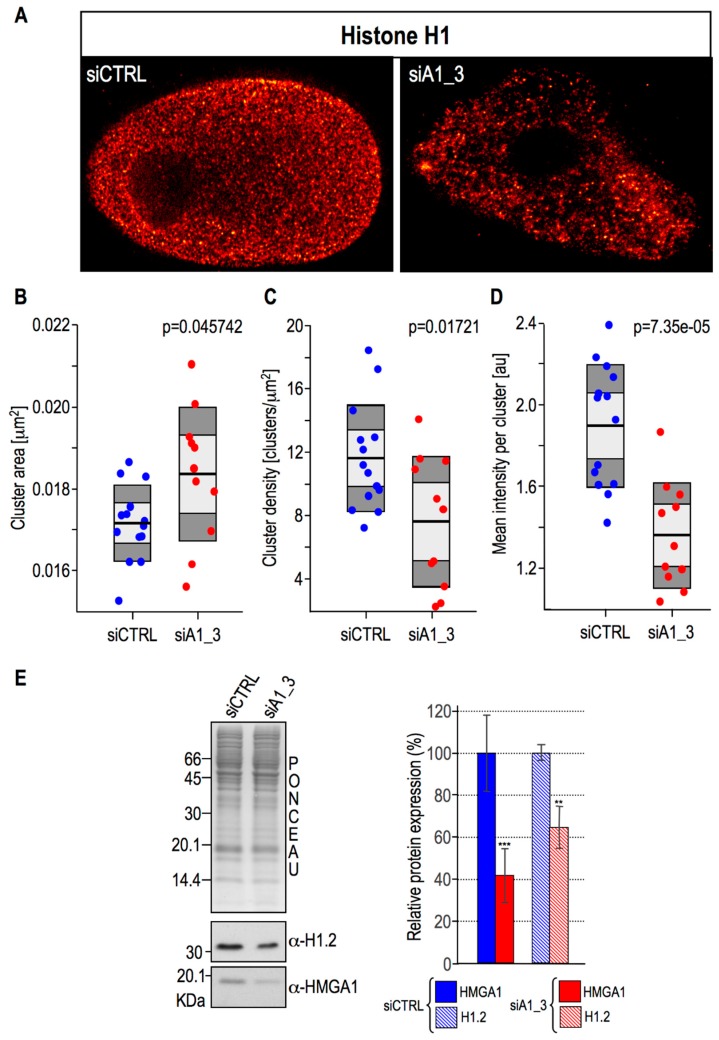
HMGA1 influences histone H1 nuclear distribution and protein expression. (**A**) Representative STED microscopy images of MDA-MB-231 cells silenced for HMGA1 (siA1_3) or treated with control siRNA (siCTRL) and immunostained for histone H1. (**B**–**D**) Box plots showing the median area, density, and mean intensity of histone H1 clusters. (siA1_3: *n* = 11; siCTRL: *n* = 14). (**E**) Western blot analyses to assess HMGA1 and histone H1.2 protein expression levels in MDA-MB-231 cells silenced for HMGA1 (siA1_3) or treated with control siRNA (siCTRL). Representative WB analyses are shown together with red ponceau stained membranes to verify total protein normalization. The histogram graphs relative to Western blot analyses were obtained using densitometric analyses (siCTRL versus siA1_3). Bars indicate the means. Standard deviations are shown (*n* = 3). Statistical significance was assessed with Student’s t test (*: *p* ≤ 0.05; **: *p* ≤ 0.01; ***: *p* ≤ 0.001). Molecular weight markers are indicated on the left (kDa).

**Figure 6 ijms-20-02733-f006:**
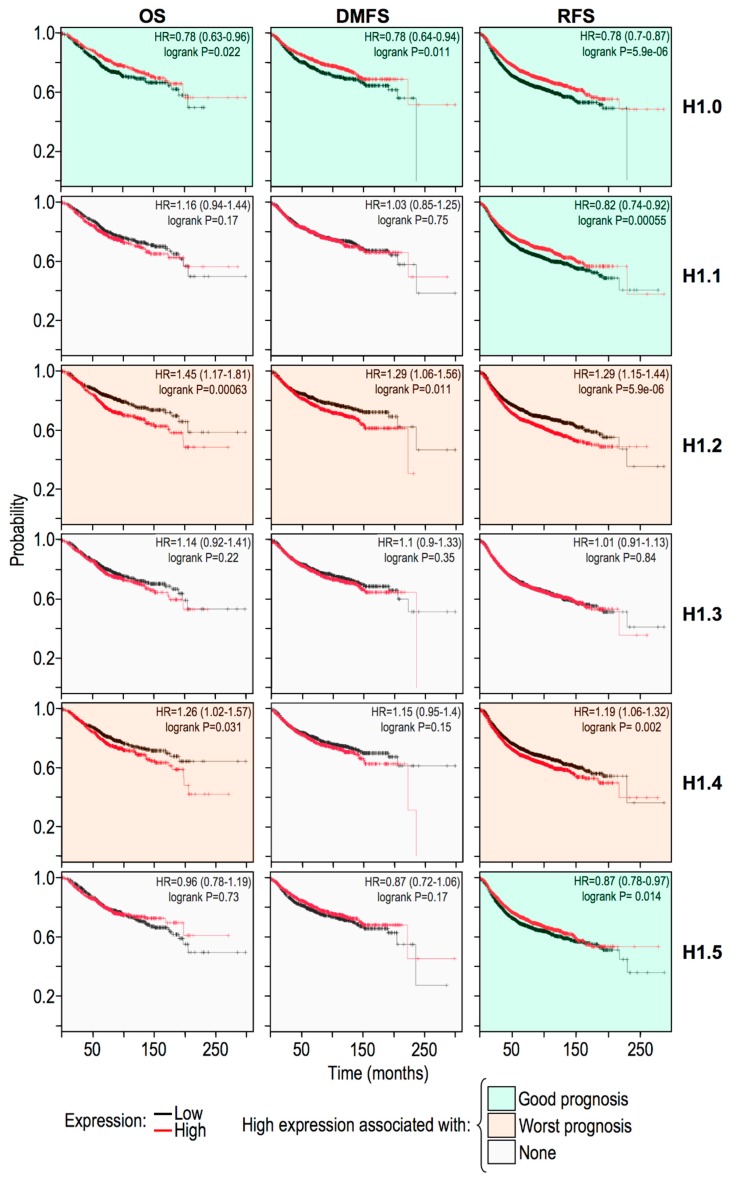
The prognostic value of histone H1 variants gene expression levels in breast cancer specimens. Kaplan–Meier plots for overall (OS), relapse-free (RFS), and distant-metastasis-free (DMFS) survival with regards to histone H1.0, H1.1, H1.2, H1.3, H1.4, and H1.5 gene expression level (low or high) in a collection of breast cancer gene expression data sets using KM plotter (kmplot.com).

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
