# Peer review of "The High Mobility Group A1 (HMGA1) Chromatin Architectural Factor Modulates Nuclear Stiffness in Breast Cancer Cells"

_ijms, 2019, doi:10.3390/ijms20112733_

Reviewer 1 Report

Major points

(1)Fig.3 and Fig.5

If the authors have the result of the same experiment using MB157 and MCF7, the conclusion is more convincing.

(2)Fig.6

Is there any difference of rate of metastasis by the type of H1?

Minor points

Fig.1

(1) A figure that shows de-phosphorylation of RSK by D1870 is needed.

(2) What are the other targets of D1870?

(3) How about the result of the same experiment using MB157?

Author Response

see the PDF file attached

Reviewer 2 Report

Article by Senigagliesi et al. presents potentially interesting results on the HMGA1, a chromatin modulator,  that might have an impact on the biophysical properties of the cells. These results may be important in the context of cancer cell metastasis. However, some concerns have arisen and should be addressed before the publication.

 Major concerns:

Fig.1. It seems that DMSO stimulate proliferation of the cells and also influences the MDA–MB–231 differentiation. How do authors explain this? Authors should perform the cytotoxicity and BRDU assays to show how DMSO and BI-D1870 affect the cell viability and proliferation of the cells. The observed effects of the BI-D1870 on the stiffness might be associated with the not optimal concentration of the DMSO . General toxicity changes the biophysical properties of the cells  e.g. PLoS One. 2012;7(5):e36885. I recommend to prepare the higher concentrated stock of the BI-D1870 to decrease the concentration of the DMSO in the cells.  

Authors should provide better picture of the cells treated with the BI-D1870. It is quite difficult to see details.

 It seems that authors used improper statistical test for the analysis of the results presented in the Fig. 1. t-student test is used for comparison of the means of two groups, here we have 3 groups and thus ANOVA with post hoc test should be performed.

Fig.5E. Authors show the histogram graphs of Western blots, however they present only the absolute values of densitometric analysis, whether these values should be standardized on the total protein control (e.g. beta actin, and it is difficult to understand why they use ponceau staining instead).

 Minor concerns:

 Fig. 1. Please indicate the concentration of DMSO.

Results section

2.2. Modulation of HMGA1 expression levels alters cellular stiffness in breast cancer cell lines.

 Lines 118-121 authors describe results from the Fig.2 not Fig.1.

 Discussion, Lines 282-283 Authors claim: “In this work, we provide evidences that the global chromatin opening (…)” Could authors describe these evidences?

 Materials and methods section

4.1. Cell Culture and Treatments – please provide the source of the cell lines, siRNA and RSK2 inhibitor BI–D1870.

 4.2. SDS PAGE and western blot analyses.

Please describe in detail the process of production of the α-HMGA1 antibody.

 Please provide the source of the HA–HMGA1 protein expression vector, empty HA–expression vector and describe the process of transfection of the MCF-7 cells. Did authors check the efficiency of transfection?

Author Response

see the attached PDF file

Round  2

Reviewer 1 Report

Minor point 1

I understand that D1870 was used as a positive control for the change of cell stiffness and cell migration, however, in that case the word "RSK2" should not be used in the manuscript since no direct experimental data is shown. In addition to the data of suppression of RSK2, higher quality journal would even ask you the result of using the kinase dead mutant or inhibitor resistant mutant.

I refrain from talking more about off-target effect because it would be too much for the authors, however, showing suppression of RSK2 by the concentration used is a minimum data if the authors would like to mention about RSK2 in the manuscript.

Figure 1 for reviewers  could suggest that D1870 activity might be related to cell motility, but not RSK2.

Reviewer 2 Report

The authors have answered my questions and I have no further comments.
